# The Influence of Iron Particles and Polyethylene Glycol on Selected Properties of Polylactide-Based Composites

**DOI:** 10.3390/polym17020146

**Published:** 2025-01-09

**Authors:** Zbigniew Oksiuta, Dominika Nowińska, Magdalena Joka Yildiz, Marek Jałbrzykowski

**Affiliations:** 1Faculty of Biomedical Engineering, Bialystok University of Technology, Wiejska 45C, 15-351 Bialystok, Poland; dominika.nowinska@sd.pb.edu.pl; 2Faculty of Civil Engineering and Environmental Sciences, Bialystok University of Technology, Wiejska 45E Street, 15-351 Bialystok, Poland; m.joka@pb.edu.pl; 3Faculty of Mechanical Engineering, Bialystok University of Technology, Wiejska 45C, 15-351 Bialystok, Poland; m.jalbrzykowski@pb.edu.pl

**Keywords:** polylactide (PLA), composite, mechanical properties, hydrolytic degradation, metallic powders, polyethylene glycol (PEG)

## Abstract

This article presents the characteristics of composites comprising polylactide combined with iron powder, from 1 to 10 wt.%, and nanoiron powders with a mass fraction from 0.1 to 1.0 wt.%, along with polyethylene glycol. A total of nine composites were prepared, with three variations each: polylactide with iron powder, polylactide with nanoiron powder, and polylactide with micro- and nanoiron powder combined with polyethylene glycol. The samples underwent mixing, extrusion, and pressing processes. To assess the properties of the resultant composite samples, ultimate tensile tests, Shore hardness tests, fracture surface observations, degradation tests in 0.9% saline solution, and DSC analyses were conducted. The findings revealed that nanoiron powder incorporated into the polylactide matrix demonstrates better tensile properties, both strength and elongation, compared to those incorporating micrometric-iron powder only. However, both iron powder additions led to a decrease in the total elongation of neat polylactide acid except for the composite with 1% nanoiron. Furthermore, all samples with polyethylene glycol addition show a lower Young’s modulus compared to neat PLA. In general, the microiron powder decreases the Young’s modulus of PLA composites, whereas the nanoiron powder slightly increases the Young’s modulus of these samples. Polyethylene glycol, a biocompatible substance, emerged as a suitable candidate for enhancing the adhesion of iron particles and improving the strength and elongation properties of the tested composites. Also, fracture surface analysis of the tensile samples suggests using fine nanoiron particles instead of coarse ones to improve the mechanical properties due to the stronger bonding of nanoiron particles to the PLA matrix.

## 1. Introduction

Biodegradable polymers such as polylactide (PLA) are currently the subject of much research work. This is due to their unique properties related to the possibility of biodegradation, as well as biocompatibility and the possibility of bioresorption. They find application in both the manufacturing of everyday items as well as in medicine, particularly in tissue engineering, regenerative medicine, and orthopedics [1]. The mechanical properties of PLA may vary depending on the molecular weight of the polymer and its degree of crystallinity. These properties can be regulated by the stereochemical configuration of the molecular structures of the PLA framework as well as the method of their production [2,3]. PLA comes in various types that are designed for specific applications such as extrusion and thermoforming, injection molding, film production, fiber manufacturing, or 3D printing. Each type is distinguished by its unique set of properties, as detailed in Table 1 [4]. In general, the semi-crystalline polylactide has a Young’s modulus (E) of about 3.5 GPa, with a maximum elongation at break of about 6%. The tensile strength of the PLA polymer without additives is from 50 to 70 MPa, the flexural modulus is ~5 GPa, the flexural strength is ~100 MPa, and the density is from 1.2 to 1.4 g/cm^3^ [5,6,7]. Often, when PLA is used for medical purposes, it is also necessary to change the degradation time of this material. The degradation rate of PLA depends on the polymer composition (PLA + additives), pH, product geometry, molecular weight of the polymer, its crystallinity, the addition of drugs and excipients, the sterilization method, and the production process [8]. Polylactide has some limitations in engineering applications, such as low ductility, which makes this polymer brittle [9].

The properties of PLA can be modified in several ways, including by adding fillers, plasticizers, stabilizers, or surface modifications [10]. The analysis of literature data shows that the influence of metallic additives, such as silver, iron, cobalt, chromium, titanium alloys, or magnesium, on PLA is increasingly being investigated [9,11,12,13,14]. The results of the research show that composites made of polylactide with metallic additives have a wide application potential. In composites of this type, the adhesion of metal particles to PLA must be favorable enough not to deteriorate the mechanical properties of such materials. Therefore, a biocompatible substance combining both substrates (PLA + metal) is being sought. Such materials may include, for example, polyethylene glycol (PEG), pluronics, or amines (e.g., polyethylamine) [15,16,17].

A very important issue regarding the medical application of biodegradable materials for bone restoration is the aspect of degradation and how this degradation takes place, whether the change in material stiffness during degradation does not negatively affect the peri-osseous areas [18]. Too rapid degradation of such material in the human body can easily cause aseptic tissue inflammation [19]. Research on the degradation process is usually carried out by assessing this phenomenon using static methods. A polymeric biodegradable sample is placed in a saline solution (SBF), sometimes with various additives, and then placed in an incubator to simulate the human body temperature of 37 °C. Such a sample is removed from the solution at specified intervals, dried, and then its various properties are checked. The medium in which the sample is located is also often examined, such as the pH, mass reduction, or electrical conductivity. Some authors [20,21] replace the fluid in which the sample is located at set intervals, e.g., every week, to better simulate the conditions prevailing in the human body.

In our previous research work [22], various additives were used to find out more about the PLA-based composites’ degradation and their mechanical properties. In this work, the mechanical, thermal, and morphological properties of composites based on polylactide with a 5% addition of polyethylene, iron, and magnesium powders were investigated. The addition of iron powder resulted in an improvement in the tensile strength (both σ and YS_0.2_) of neat PLA, while the magnesium powder had a positive impact on the ductility of the composite material (ε = 2.5 ± 0.76%). The iron-doped composite proved to be the most stable of all tested materials during the degradation test in 0.9% saline solution. In this work, iron was chosen as the most promising material. The main goal of this article is to investigate different iron powder morphologies and contents on the PLA degradation and mechanical properties of such composites. Because the preliminary results revealed that the mechanical properties of composites with iron powders deteriorated, PEG was added to the PLA-Fe-based materials. PEG was added in the form of a powder (with a molecular weight of 3000) and liquid form (with a molecular weight of 300) to check the effect of this compatibilizer on the properties of the composite samples.

## 2. Materials and Methods

### 2.1. Samples Preparation

Nine different compositions based on PLA 3001D (NatureWorks, Blair, NE, USA) were prepared with the addition of Fe, nanoFe, and PEG in liquid and powder form (Sigma-Aldrich, Saint Louis, MO, USA). The powder particle size of iron was 45 µm (Johnson Matthey, London, UK), and for iron nanopowder, spherical particles with size 60–80 nm were used. Both powders have a spherical shape, as shown in Figure 1. Compositions of PLA-based composites with mass concentrations and their densities are shown in Table 2.

The samples were prepared by mixing all components in a crusher and then extruding them in an EHP 25 Eline laboratory extruder (Mercator, Skawina, Poland) with the following process parameters: the temperature of the hopper zone, 50 °C; the temperature of the supply zone, 170 °C; the temperature of the compression zone, 180 °C; and the temperature of the dispensing zone, 180 °C. The extruder head temperature was 175 °C, and the extrusion speed was set at 5% of the maximum machine speed. Immediately after extrusion, the extruded composite material was cut to a length of 100 mm and then pressed under a pressure of 100 MPa using a hydraulic press to obtain a uniform thickness of 2.5 mm. The tensile samples were mechanically shaped by milling and grinding. Five tensile specimens of each composition were tested. The prepared materials and sample dimensions are shown in Figure 2.

### 2.2. Hardness, Tensile Tests, and Microstructure Observations

Hardness tests were carried out using a Shore durometer—the Zwick Roell 7206 D type (Zwick Roell, Ulm, Germany). Tensile tests were carried out using an MTS Insight testing machine with an extensometer (measuring base of 25 mm). The tests were carried out with a constant cross-head displacement rate of 2 mm/min at room temperature. An example of the tensile stand is shown in Figure 3.

The surface of the samples was observed by confocal microscope using the 3D Measuring Laser Microscope LEXT OLS4000 (Olympus, Shinjuku, Tokyo, Japan).

### 2.3. Density and Weight Loss Tests

The degradation process of the PLA-based composites was followed by determining the weight loss of the samples. The samples with a size of 10 × 10 × 4 mm^3^ were placed in 0.9% saline solution (NaCl) and stored in an incubator at 37 ± 0.5 °C. At an established time, the samples were removed and dried, and the weight of the samples was then checked. The percentage of weight loss was determined by comparing the dry weight (*w_d_*) with the initial weight (*w*_0_) according to Equation (1):(1)$$weight\;loss\;\left(\%\right)=\frac{w_{0}-w_{d}}{w_{0}}\times 100\%$$

For density and weight loss measurements, a balance (Mettler Toledo, Columbus, OH, USA) with a sensitivity of 0.01 mg was used. The values of pH and conductivity for the solution were also checked. Three to five measurements were made, the average was taken, and the standard deviation was calculated. Measurements were made using the METTLER TOLEDO XS205 SevenMulti (Mattler-Toledo, Warsaw, Poland).

### 2.4. Differential Scanning Calorimetry

Composite samples with a mass of ca. 7.0 mg were unmolded from the filament using an iron blade. The samples were then weighed by a Mettler Toledo XS204 (Mattler-Toledo, Poleczki, Warsaw, Poland) series balance having an accuracy of 0.1 mg.

Differential scanning calorimetry (DSC) experiments were conducted with a Q2000 DSC TA Instrument by placing the polymer samples in a Tzero aluminum crucible and holding them in an inert nitrogen (purity 99.999%) atmosphere isothermally at 20 °C for 5 min. Following that, the samples were heated with a heating rate of 10 °C/min to 200 °C (ca. 30 °C above the melting point of 170 °C). Then, through opening the measuring cell, the samples were rapidly quenched by being placed on a piece of metal having a temperature of 5 °C. After quenching, the material was held isothermally at 20 °C for 5 min. Subsequently, the samples were heated with a heating rate of 10 °C/min to 100 °C. The cold crystallization, melting enthalpies and melting temperature (*Tm*), and glass transition (*Tg*) characteristic temperatures (onset, midpoint, and heat capacity change (Δ*Cp*)) were obtained using the Universal Analyses 2000 software (TA Instruments—Waters LLC, New Castle, DE, USA). The glass transition (*Tg*) characteristic temperatures (onset, midpoint, and heat capacity change (Δ*Cp*)) were obtained from the second heating. The degree of crystallinity (*X_c_*) was estimated using the following Equation (2):(2)$$X_{c}=\frac{∆H_{m}-∆H_{c}}{∆H_{100\%}}\times 100\%$$
where Δ*H*_100%_ = 93.7 J·g^−1^ according to [23]; Δ*H_c_* is the enthalpy of cold crystallization; and Δ*H_m_* is the enthalpy of melting. The degree of crystallinity was corrected for the filler content.

### 2.5. Physicochemical Properties in Saline Solution

Weight loss, pH, and electrical conductivity tests of the PLA composite samples incubated at 37 °C for 180 in 0.9% NaCl solution were performed and analyzed. Initially, these parameters were tested every 3 days for a period of 28 days, then after 35, 42, 50, 90, 120, 150, and 180 days, respectively. pH and conductivity tests were performed using the appropriate electrodes in conjunction with a Seven-Multi multifunctional ionic conductivity meter (Mettler Toledo; Greifensee, Switzerland) (Mettler Toledo; Greifensee, Switzerland). pH ratings were performed using a Clarytrode 120 electrode (Mettler Toledo, Columbus, OH, USA). Electrical conductivity was carried out using an InLab740 conductivity cell (Mettler Toledo, Columbus, OH, USA)) with an integrated probe to measure the temperature of the tested solution. The mass changes of the PLA-based composite samples in air and in 0.9% NaCl solution was recorded by the balance (Mettler Toledo, Columbus, OH, USA) using Archimedes’ principle. Measurements were performed three times for each sample. The mean values and standard deviation were calculated. All tests were executed at the temperature of 21 ± 1 °C.

## 3. Results and Discussion

### 3.1. Tensile and Hardness Properties

Table 3 summarizes the results of the Shore hardness test and tensile properties for all tested composites. Figure 4 shows the stress–strain curves for the tested composites with the PEG additive in powder and liquid forms, where the best results are visible for a composite based on the PLA with an addition of 1% nanoFe and 1% PEG in a solid state (powder).

In general, the highest tensile strength and elongation values for composites without the PEG addition were obtained for the material with 1% nanoFe. In comparison with pure PLA, the tensile strength increased by ~15%, whereas elongation increased by ~40%. Note that both the strength and elongation of these composites increase with the increase in the content of nanoFe powder, from 0.1 wt.% up to 1.0 wt.%. The same trend is found in the case of the addition of the iron micrometric powder, increasing from 1.0 wt.% to 10 wt.% of Fe.

The addition of iron and PEG powders caused the greatest reduction in the Young’s modulus of the tested samples. Conversely, the incorporation of nanoFe led to a slight increase in Young’s modulus compared to the neat PLA. The Young’s modulus of the PLA + Fe composites decreases with increasing Fe powder content. Generally, adding microFe powder decreases the Young’s modulus of PLA, while adding nanoFe powder increases the Young’s modulus values. NanoFe powder, compared to microFe powder (see Figure 4), is finer and has a high surface-to-volume ratio, which allows for stronger attachment to PLA. The rounded shape of the microiron particles makes it more difficult to form a permanent bond with polylactide. The properties mentioned above influence the mechanical properties of the composite.

The addition of PEG in the form of powder (3000 molecular weight) significantly improves the properties of the PLA composites, while PEG in the form of a liquid decreases the tensile properties compared to the similar PLA composition without the addition of PEG. However, it can be concluded that PEG has an overall beneficial effect on increasing the elongation of PLA composites.

With increasing Fe and nanoFe content, the hardness of the samples constantly decreases, and the hardness of almost all composite samples (except PLA + 1%Fe) has the lowest value in comparison to the neat PLA (PLA 100%).

For the composites with PEG, the highest tensile strength was achieved for the PLA with 1% nanoFe and 1% PEG in the form of liquid. This composite also achieved the highest values of tensile strain and hardness.

### 3.2. Mass, pH, and Conductivity Changes

Changes in the mass of the tested samples are shown in Figure 5. The samples were kept in the 0.9% saline solution for up to 180 days. It can be seen that the mass of the samples first increased, then gradually decreased.

In the initial stage of the process, up to 15 days, an increase in the weight of the samples is apparent, which may be due to the swelling of the hydrophilic PLA. Then, the mass decreases, and after 50 days, it remains at an almost similar level until the test is interrupted, as evidenced by the decreasing scatter of the obtained results (see Figure 5). Note that the mass change is not spectacular and ranges from 98–102%. These results also indicate that after 180 days of exposure, the highest mass loss possesses the material with the lowest PLA + 0.1% nanoFe content, with the lowest initial crystallinity of ~24% (see Table 4), while in the material with the highest Fe content PLA + 10% Fe and PLA + 1%nanoFe, with the highest crystallinity, the mass degradation is the lowest.

It is known that the polylactide undergoes hydrolytic degradation in an aqueous environment. This proceeds by random cleavage of the ester bond, which is controlled by the random cleavage of the ester chain, the amount of water absorbed, the diffusion coefficient of the chain and its chains within the polymer, and the solubility of the degradation products [24,25,26]. This phenomenon can be explained as follows. The amorphous layer is usually formed in the PLA from the outside due to the fastest heat transfer after extrusion or injection. Therefore, the crystalline part of the sample is located in the core of the sample. The amorphous layer readily absorbs water, which causes the mass gain. This leads to chain scission, reducing the number of branches and the molecular weight of the polymer, which in turn becomes more susceptible to the degradation process. As a result, it begins to degrade, and the sample mass decreases. Then, the water penetrates into the core, where the crystalline structure dominates, of which the partial hydrolysis again affects the mass of polylactid acid. The final products of PLA decomposition are water and carbon dioxide.

Figure 6 shows the change in the pH of the 0.9% saline solution in which the composite samples were kept for 180 days at 37 ± 0.5 °C. At the beginning of the process, the pH of the liquid oscillated around 7.0. A gradual increase in pH can be seen for all samples with the iron and nanoFe additions to about pH 8.5. After approximately day 80, a drop in this parameter to a value of ~7.5 is visible for all tested samples. The drop in pH is probably caused by the release of H+ ions into the solution, which is produced during the decomposition of PLA. The solution penetrating the composite causes corrosion of the iron particles inside, and the resulting rust is a mixture of iron hydroxides, oxides, and carbonates, which may cause an alkaline reaction in the solution.

In Figure 7, changes in the conductivity of liquid are presented. It can be seen that the conductivity remains at the same level (13.5 mS/cm) until about day 50, then increases, reaching a maximum value of 15.5 mS/cm around day 80. Subsequently, the conductivity drops significantly below the initial value on day 110 to even about 8 mS/cm after 180 days of testing. This happens for both the iron- and nanoiron materials.

Note that changes in the mass, pH, and conductivity are correlated. During the initial phases of the study, conductivity remains relatively stable for up to 80 days, at which point it starts to escalate rapidly. This is caused by the release of iron ions, which increase the conductivity of the solution. Also, the pH changes show their peak value after 80 days of incubation, followed by a gradual decline, likely attributed to the decomposition of PLA and the subsequent release of H+ ions, thereby acidifying the environment. The presence of PLA in the solution causes a decrease in conductivity and a slow decrease in the sample mass.

### 3.3. DSC Test Results

The results obtained from the DSC measurements and calculations based on the obtained data are presented in Table 4 and Table 5 and Figure 8 and Figure 9, respectively.

The degree of crystallinity of pure PLA indicates that the material, after extrusion and pressing, has an 82.4% amorphous structure, and the degree of crystallinity is much lower than for the composite samples. The increase in crystallinity is obtained with the increase in the content of microFe and nanoFe powders. The addition of nanoFe powder resulted in a significant increase in crystallinity compared to the microFe powder, which emphasizes the role of nanoFe particles in this parameter. However, between 0.5 and 1.0% of nanoFe, the difference in crystallinity is less than 3%. It should be noted that the crystallinity of the microFe powder samples clearly correlates with the mechanical properties of these composites; however, better results were obtained for the neat PLA. The above-mentioned properties of the microFe powder composites do not differ much when considering the upper and lower ranges of these values, which may indicate a slight influence of microFe powder on the mechanical properties of the composites. However, in the case of nanoFe powder, the differences are significant. This indicates that a different mechanism operates during the extrusion of the material when micro- and nanofillers are present in its structure. In each case, the structure of PLA is altered, and the course of its ordering is different during the transition from the plasticized to the glass transition state.

The glass transition temperature (*Tg*) analysis of the PLA and its composites with iron and nanoiron powders reveals notable shifts in thermal behavior depending on the type and concentration of the filler material (Table 4).

Pure PLA exhibits a *Tg* midpoint at 50.21 °C with a heat capacity change (Δ*Cp*) of 0.7318 J/(°C·g). However, adding iron powder significantly increases the *Tg*, with the midpoint values ranging from 66.32 °C to 68.30 °C, depending on the concentration of Fe powder. The increase in *Tg* suggests that the Fe powder restricts the mobility of the polymer chains, leading to a stiffer composite structure. Moreover, the highest Δ*Cp* (1.1420 J/(°C·g)) is observed in the PLA with 5% Fe powder, indicating that this composition induces more significant changes in the amorphous region of the polymer [27,28,29,30,31].

For the nanoFe composites, the *Tg* is generally lower than for the microFe composites, with midpoint values ranging from 61.97 °C to 63.18 °C. This lower *Tg* compared to the microFe indicates that nanoFe particles, while still increasing the *Tg* relative to pure PLA, exert less restriction on polymer chain mobility compared to their micro-sized counterparts. Additionally, the Δ*Cp* values for the nanoFe composites are lower than both the neat PLA and microFe composites, which could imply a less effective filler–matrix interaction or a reduction in the overall amorphous content.

The DSC analysis of the PLA and its composites with Fe and nanoFe powders reveals distinct differences in the thermal properties, particularly in cold crystallization enthalpy (Δ*Hc*), melting temperature (*Tm*), melting enthalpy (Δ*Hm*), and degree of crystallinity (*Xc*). Pure PLA exhibits the lowest cold crystallization enthalpy (3.42 J/g) and a melting temperature of 153.25 °C, indicating lower energy requirements for crystallization and melting compared to the composites. The addition of iron powder significantly increases both Δ*Hc* and *Tm*, with PLA + 5% Fe showing the highest Δ*Hc* (30.39 J/g) and *Tm* (171.95 °C), reflecting greater disruption in polymer chain mobility and higher thermal stability. In contrast, nanoFe-filled PLA composites demonstrate lower Δ*Hc* values, with the 1% nanoFe composite showing a cold crystallization enthalpy of 12.49 J/g, indicating less interference in the crystallization process compared to the microFe composites.

Additionally, the degree of crystallinity is significantly reduced in the PLA with iron particles. PLA + 1% Fe shows the lowest *Xc* (20.71%), while the nanoFe composites have a degree of crystallinity ranging from 25.02 to 44.89%, with the highest value for the 1% nanoFe sample. These results suggest that both powders significantly alter the thermal behavior of neat PLA; however, it seems that the microFe powders appear to have a more moderate effect on crystallinity.

In the context of PLA, the formation of different crystal forms, specifically the α and α’ crystals, is influenced by the thermal history and processing conditions of the material [32]. The α form is typically more thermodynamically stable and forms at higher crystallization temperatures, while the α’ form is less stable and forms at lower temperatures. In the present study, the DSC data suggest that the incorporation of iron and nanoiron powders affects the crystallization behavior and the potential formation of these crystal forms. The increase in cold crystallization enthalpy and melting temperature observed in the PLA composites with iron powders, especially at higher concentrations, may indicate a shift towards more stable α crystals, as higher temperatures favor the formation of this crystalline structure. In contrast, the nanoiron composites, which exhibit lower cold crystallization enthalpy and a more moderate increase in melting temperature, suggest a more heterogeneous crystal structure, potentially involving a mixture of α and α’ crystals.

Table 5 and Figure 9 show the thermal characteristics of PLA-based composites with the addition of Fe and nanoFe particles and PEG.

The glass transition temperatures of the PLA composites were influenced by the addition of iron and polyethylene glycol (PEG) in both powder and fluid forms (Table 5). The pure PLA sample exhibited a *Tg* onset at 45.24 °C, with a midpoint of 50.21 °C and a ΔCp of 0.7318 J·(°C·g)^−^^1^. The addition of 1% nanoFe and 1% PEG in powder form shifted the *Tg* to a higher temperature, with an onset at 57.64 °C and a midpoint of 59.45 °C, while the Δ*Cp* decreased to 0.4681 J·(°C·g)^−^^1^, suggesting restricted chain mobility. When PEG was introduced in fluid form at the same concentration, the *Tg* decreased slightly, with an onset of 49.70 °C and a midpoint of 51.65 °C, accompanied by a higher Δ*Cp* of 1.300 J·(°C·g)^−^^1^, indicating enhanced chain flexibility. For the sample with 10% Fe and 10% PEG (fluid), the *Tg* onset and midpoint were 50.86 °C and 54.35 °C, respectively, with a notably high Δ*Cp* of 1.971 J·(°C·g)^−^^1^, reflecting significant changes in the polymer matrix due to the increased additive content.

The differential scanning calorimetry (DSC) results provide insights into the crystallization behavior, melting temperature, and crystallinity of the PLA composites. The neat PLA sample showed an exothermic crystallization enthalpy of 10.66 J·g^−^^1^, a melting temperature of 153.25 °C, and a melting enthalpy of 19.91 J·g^−^^1^, with the lowest crystallinity of 17.60%. Upon the addition of 1% nanoFe and 1% PEG in powder form, the crystallization enthalpy slightly increased to 11.17 J·g^−^^1^, with a significant rise in melting temperature to 167.52 °C and a corresponding increase in melting enthalpy to 56.12 J·g^−^^1^. However, the degree of crystallinity increased significantly to 47.97%, indicating major changes in the ordered structure. For the sample with 1% nanoFe and 1% PEG in fluid form, the crystallization enthalpy increased markedly to 19.08 J·g^−^^1^, suggesting enhanced crystallization during cooling. The melting temperature was measured at 164.22 °C, while the melting enthalpy reached 55.36 J·g^−^^1^. Also, the crystallinity was lower than that of the PEG sample in powder form, at 38.72%; however, this was more than twice as much as the neat PLA. In the composite with 10% Fe and 20% PEG (fluid), the crystallization enthalpy decreased to 17.70 J·g^−^^1^, with a melting temperature of 164.89 °C and the highest melting enthalpy of 57.29 J·g^−^^1^. The crystallinity was slightly reduced to 42.25%, likely due to the higher content of additives interfering with the PLA’s crystalline arrangement. These results suggest that the type and amount of PEG and iron content and size influence both the crystallization and melting behaviors of PLA composites, with higher additive concentrations leading to increased crystallinity.

Upon analyzing the mechanical properties alongside the DSC data, it has been observed that the UTS generally decreases with increasing iron content in PLA composites. For example, PLA + 10% Fe shows a reduction in UTS (29.5 MPa) compared to pure PLA (36.0 MPa), which correlates with the reduction in the degree of crystallinity for this composite (29.00% compared to 44.66% in pure PLA). This suggests that the lower crystallinity of the iron-filled PLA composites leads to weaker material, as less crystalline material means a reduced ability to withstand tensile forces. However, PLA + 1% nanoFe and PLA + 1% nanoFe + 1% PEG (fluid) show higher UTSs (40.6 MPa and 45.88 MPa, respectively) despite maintaining moderate levels of crystallinity (44.64% and lower), indicating that the nano-sized iron and PEG additions enhance the tensile strength, likely due to a better filler–matrix interaction and more effective load transfer. Moreover, in the case of nanoiron- and PEG-containing composites, PLA + 1% nanoFe + 1% PEG (fluid) exhibits a relatively high hardness (67.5 Shore D) despite moderate crystallinity, implying that the presence of nanoiron and PEG improves surface rigidity without a significant impact on crystallinity.

### 3.4. Microscopic Observations

Figure 10 shows the sample surfaces observed using a confocal microscope. These images show PLA composites with the addition of iron, nanoiron powders, and PEG (fluid and powder). The images show agglomerations of iron particles (white dots), as well as individual particles located in the polylactide matrix. In certain areas, it is evident that these particles exhibit poor bonding with the PLA in images lacking PEG.

Microscopic observations of the PLA composite samples show that in the material containing 10% Fe powder, the addition of Fe in the PLA matrix was homogeneously distributed. This is in accordance with the mechanical properties data presented in Table 3 for these specimens. In the case of the PLA with 10% Fe and 20% PEG (fluid), the Fe powder particles are not clearly visible in the PLA + PEG (fluid) matrix. On the contrary, a band-like microstructure of the PLA + 1% nanoFe + 1% PEG (powder) can be observed. This band-like microstructure is likely responsible for the poorer mechanical properties measured during the tensile tests compared to the PLA + 1% nanoFe + 1% PEG (liquid) sample (Figure 10e). Figure 10e also shows an area with heterogeneous iron particle distribution. The parallel traces visible in the photos are probably the result of the orientation of the material structure after extrusion. The visible bidirectionality may result from the sample flowing under the press.

The fracture surfaces of the tested specimens were also analyzed after tensile testing. Figure 11 and Figure 12 show SEM images with low and high magnifications. Before the SEM observations, the PLA specimens were sputtered with gold to a thickness of 6 nm.

The SEM observations of all tested composites confirm the tensile strength data typical of the brittle fracture mode, including pure PLA (without a yield point and necking). The surface of pure PLA was flat with jags and twisted PLA threads (Figure 11f). Observations of the fracture of the samples at a low magnification of 50× (Figure 11, in the upper left-hand corner windows) revealed differences between each material group. Samples with the iron powder addition (from 1% to 10 wt.%) have a similar fracture mode (see Figure 11a,b, and Figure 12). The fracture surface of these samples shows a more distinct central region and fracture edges located below.

This type of fracture is caused by complex stress concentration in the samples that occurred during the tensile test. Additionally, areas covered with scales and river patterns are visible, especially in the central part of these samples. A similar fracture mode at low magnifications was also observed for the PLA + 1%nanoFe + 1%PEG (liquid) sample (Figure 11c). At higher magnification, tears of PEG are visible. On the contrary, PLA + 10% Fe + 20% PEG (liquid) and PLA + 1% nanoFe + 1% PEG (powder), shown in Figure 11d,e, have a generally smooth fracture surface with delamination and fiber-like texture, respectively. However, both materials present a completely different fracture surface phenomenon. The textured fracture surface is responsible for the highest ductility in this type of composite material. This highlights that PEG in the form of powder (3000 molecular weight) provides better tensile properties than the PEG in fluid form and will be selected for further study.

It should be noted that with the increase in Fe powder content, the concentration of observed particles also increases, and in the case of PLA with 10% Fe (Figure 12), the image shows fine, spherical iron particles of a size ~10 μm, confirmed by SEM-EDS, embedded in the PLA matrix, and with recesses of larger Fe particles (~50 μm). Observations show that more particles, especially larger ones, were pulled out than remained in the PLA matrix. This indicates that the fine Fe-powder particles have better interfacial adhesion to the PLA matrix and were not pulled out during the tensile test. This also suggests that selecting only a fine fraction of iron particles can improve the mechanical properties of the PLA + Fe composite. Moreover, this observation can also explain why the material with the addition of nanoFe powder obtained higher tensile properties.

Another aspect is the sphericity of the Fe particles. Increasing the surface area of iron particles towards using powder with a more developed surface area may also affect the binding mechanism of the Fe powder to the PLA matrix. This requires further research with powder particles obtained by methods other than gas atomization.

## 4. Conclusions

In the case of composites containing PLA and iron powders, it can be seen that the materials to which nanoFe was added have better properties than those of iron powder. This relationship allowed the selection of a nanoFe powder to be a material to which PEG was also added. PEG powder with a molecular weight of 3000 provides better tensile properties for the PLA and iron-powder composite compared to liquid PEG with a molecular weight of 600.

Among all the composites made, the best mechanical properties, both stress at break and elongation to fracture, were composites based on polylactide with additions of nanoFe powders and fluid PEG. On this basis, it can be concluded that this composite is promising, while the material with the same composition and with the powdery PEG addition significantly deteriorates the mechanical properties of such a composite. It can be noticed that PEG significantly improves the adhesion of iron particles to the polylactide matrix.

When it comes to the degradation of the made composites, similar changes can be observed in the mass of the samples, pH, and conductivity of the solution. Initially, the sample mass experiences an upward trend, followed by a decline, and then stabilizes around the 80th day. The pH of the solution initially rises, then starts to decrease after the 80th day. Similarly, conductivity maintains a steady level until the 80th day, after which it rises before declining. This is related to the mechanisms of material degradation. At first, water seeps into the material, causing it to increase in weight, and the iron begins to rust, producing substances that cause the pH to change to alkaline. Then, after 80 days, positive hydrogen ions begin to be released, causing acidification of the environment, a decrease in sample mass, and an increase in conductivity associated with the release of iron ions.

The degree of crystallinity is also influenced by the size of the Fe particles and the molecular weight of PEG. Increasing the content of Fe particles from 1% to 10% allowed for obtaining a larger amount of the amorphous phase in PLA, lowering the degree of crystallinity. The powdery PEG led to a higher degree of crystallinity compared to the liquid PEG.

Also, the introduction of Fe and nanoFe particles and PEG into the PLA matrix increased the melting temperature by approximately 10%. However, double melting was not observed for the sample containing 10% Fe, suggesting that this phenomenon can only be observed for a certain range of Fe content.

The fracture surface analysis of the tensile samples suggests using fine iron particles instead of coarse ones to improve mechanical properties due to the stronger bonding of the fine particles to the PLA matrix.

## Figures and Tables

**Figure 1 polymers-17-00146-f001:**
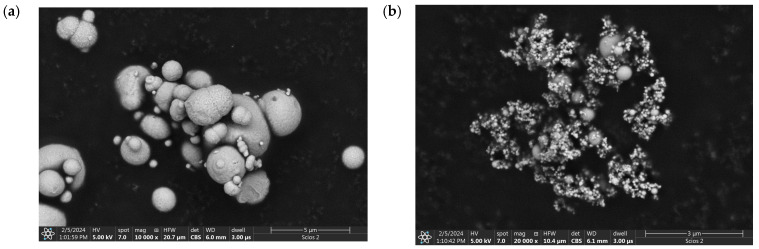
Scanning electron microscopy images: (**a**) iron particles and (**b**) nanoiron particles.

**Figure 2 polymers-17-00146-f002:**
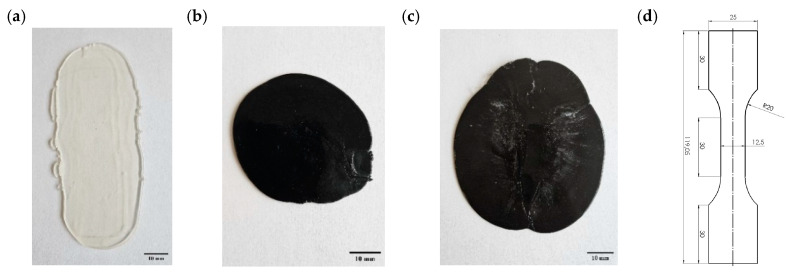
Obtained materials after extrusion and pressing: (**a**) pure PLA; (**b**) PLA + 1% nanoFe + 1% PEG (fluid); (**c**) PLA + 10% Fe; and (**d**) the final shape of the sample with dimensions selected for tensile tests.

**Figure 3 polymers-17-00146-f003:**
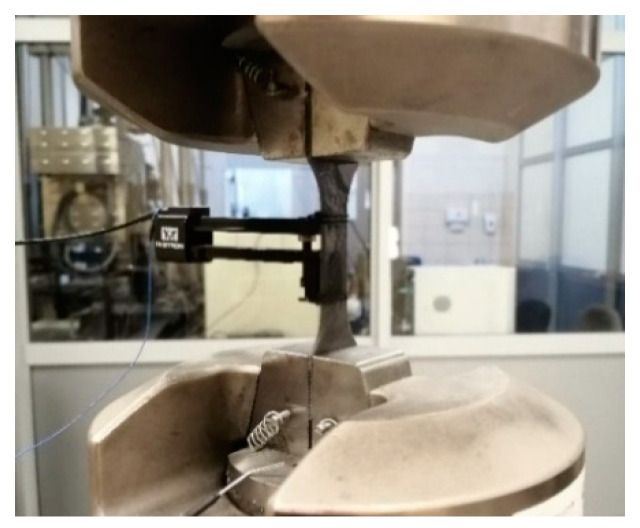
The stand test facility with a tested sample.

**Figure 4 polymers-17-00146-f004:**
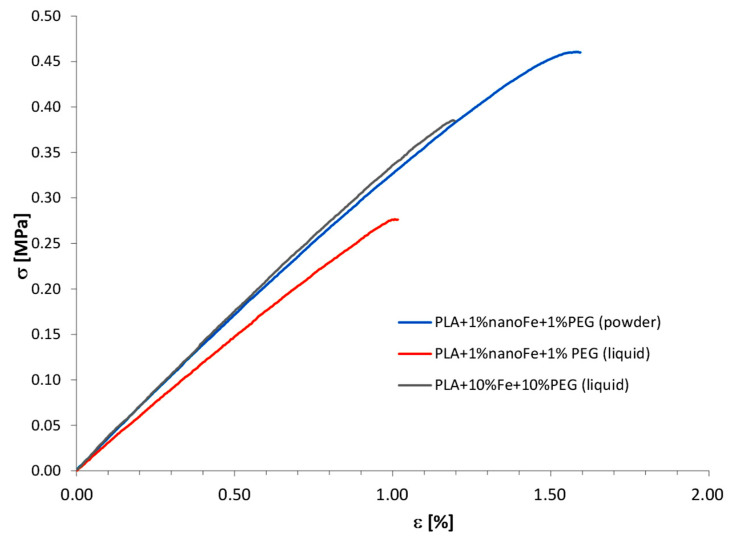
Stress–strain representative curves for the tested materials with PEG additives.

**Figure 5 polymers-17-00146-f005:**
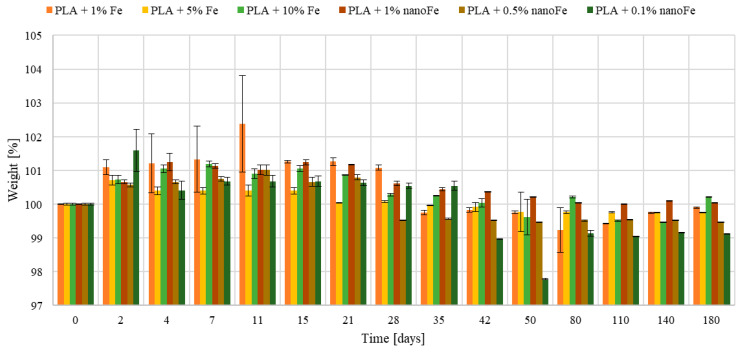
Change in mass of the tested PLA-based composite samples.

**Figure 6 polymers-17-00146-f006:**
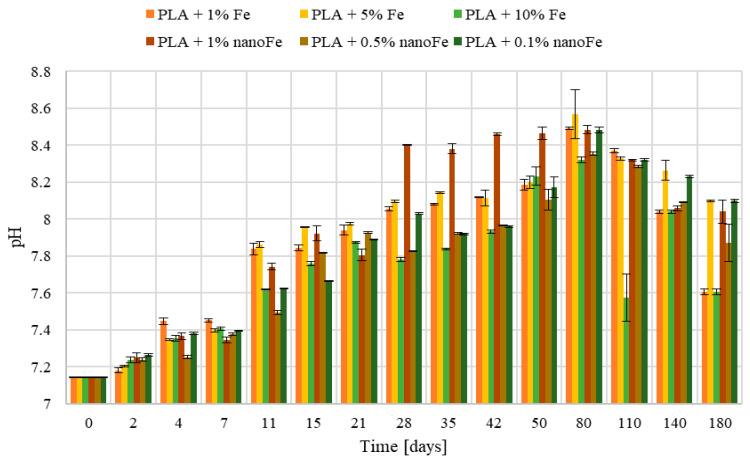
Changes in pH at different storage times in the 0.9% saline solution at T = 37 ± 0.5 °C.

**Figure 7 polymers-17-00146-f007:**
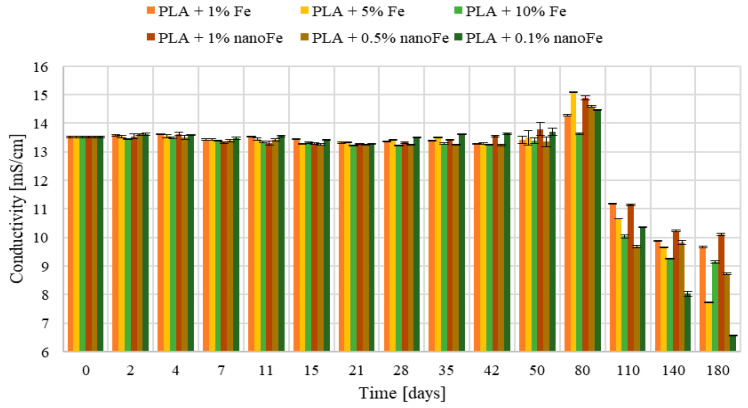
Change in conductivity at different times of process in the physiological liquid at T = 37 °C.

**Figure 8 polymers-17-00146-f008:**
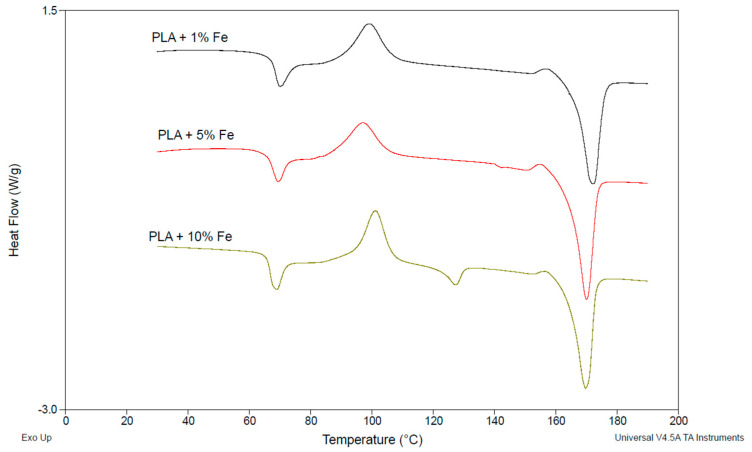
Examples of thermograms of the PLA-based samples during first heating: with 1%, 5%, and 10% of Fe (in wt.%).

**Figure 9 polymers-17-00146-f009:**
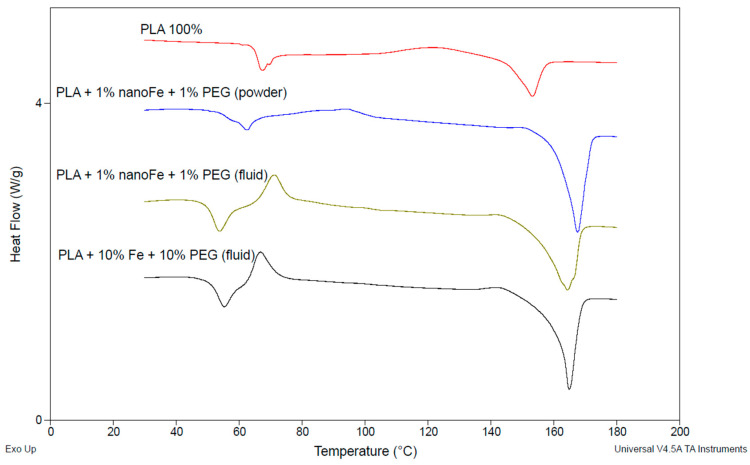
Thermograms of the PLA-based samples during first heating.

**Figure 10 polymers-17-00146-f010:**
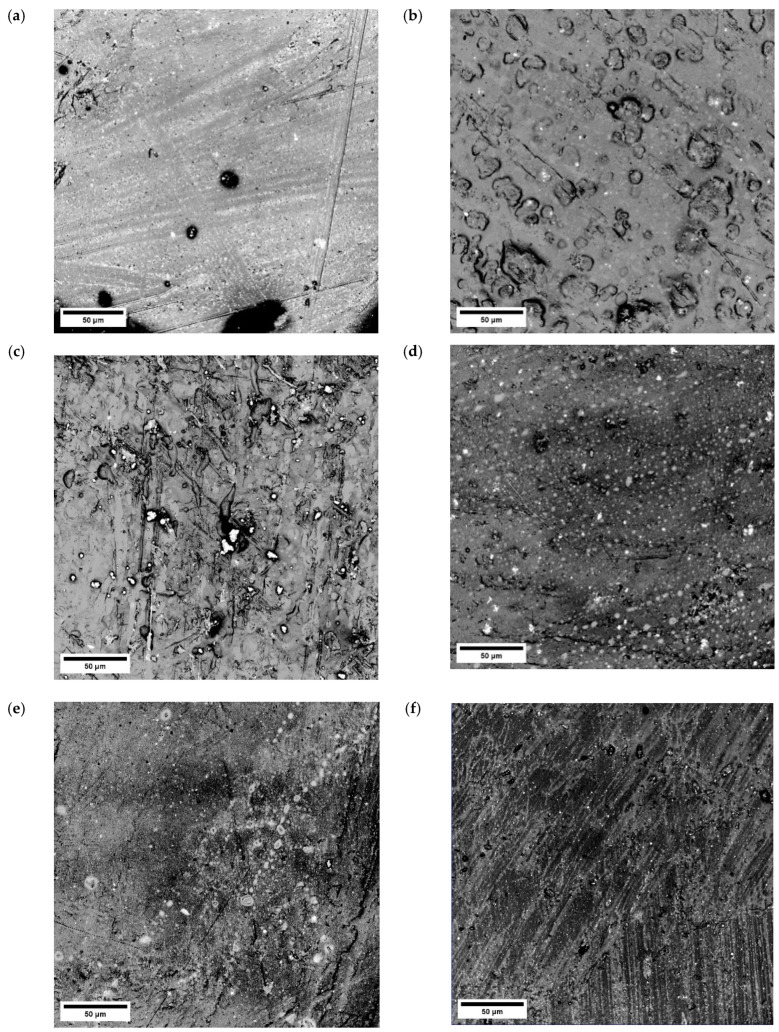
Confocal microscope images for sample surfaces (mag. 1077×): (**a**) PLA + 1% Fe; (**b**) PLA + 5% Fe; (**c**) PLA + 10% Fe; (**d**) PLA + 1% nanoFe + 1% PEG (powder); (**e**) PLA + 1% nanoFe + 1% PEG (fluid); and (**f**) PLA + 10% nanoFe + 20% PEG (fluid).

**Figure 11 polymers-17-00146-f011:**
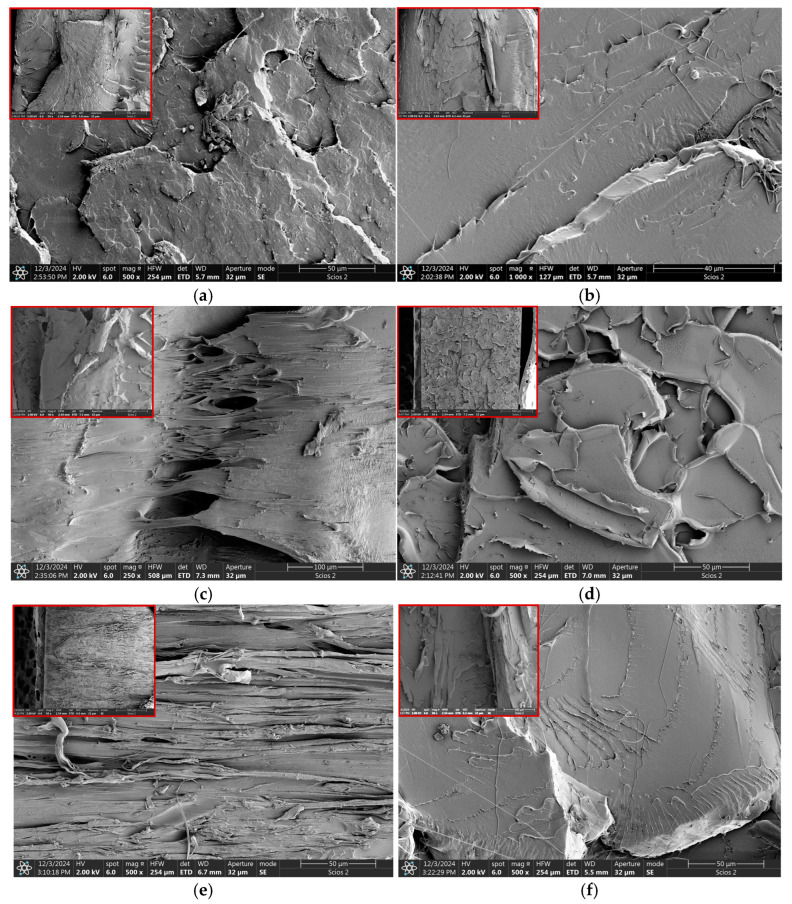
SEM images of sample fractions after tensile testing: (**a**) PLA + 1%Fe; (**b**) PLA + 5%Fe; (**c**) PLA + 1% nanoFe + 1% PEG (fluid); (**d**) PLA + 10% Fe + 20% PEG (fluid); (**e**) PLA +1% nanoFe + 1%PEG (powder); and (**f**) pure PLA. In the upper left-hand corner of the photos are SEM images at 50× magnification.

**Figure 12 polymers-17-00146-f012:**
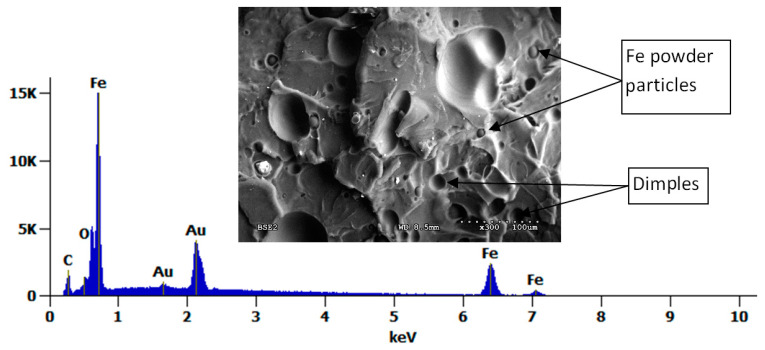
SEM-EDS analysis of PLA + 10% Fe (BSE detector).

**Table 1 polymers-17-00146-t001:** Polylactide acid properties [4].

	Material		
	PLA 3251D	PLA 3001D	PLA 4043D
Tensile Yield Strength [MPa]	62	48	60
Tensile Elongation [%]	3.5	2.5	6
Young Modulus [GPa]	3.5	3.6	3.6
Heat Distortion Temperature [°C]	55	55	55
Melt Flow Rate	80	10–30	6

**Table 2 polymers-17-00146-t002:** Bulk chemical compositions and densities of tested samples.

Composite Number	Composition/Indication	Density [g/cm^3^]
1	Polylactide (99%), iron powder (1% *)/(PLA + 1% Fe)	1.31
2	Polylactide (95%), iron powder (5%)/(PLA + 5% Fe)	1.62
3	Polylactide (90%), iron powder (10%)/(PLA + 10% Fe)	1.90
4	Polylactide (99%), iron nanopowder (1%)/(PLA + 1% nanoFe)	1.30
5	Polylactide (99,5%), iron nanopowder (0.5%)/(PLA + 0.5% nanoFe)	1.27
6	Polylactide (99,9%), iron nanopowder (0.1%)/(PLA + 0.1% nanoFe)	1.25
7	Polylactide, iron nanopowder (1%), polyethylene glycol (1%) (powder)/(PLA + 1% nanoFe + 1% PEG (powder))	1.23
8	Polylactide, iron nanopowder (1%), polyethylene glycol (1%) (fluid)/(PLA + 1% nanoFe + 1% PEG (fluid))	1.23
9	Polylactide, iron nanopowder (10%), polyethylene glycol (20%) (fluid)/(PLA + 10% nanoFe + 20% PEG (fluid))	1.14

* (in wt. %); density of 100% PLA 1.24 g/cm^3^ [4].

**Table 3 polymers-17-00146-t003:** Hardness and strength test results.

Material	σ [MPa]	Strain ε [%]	Young Modulus E [MPa]	Hardness [°Sh]
PLA 100%	36.0 ± 2.5	1.03 ± 0.05	3610 ± 76	70.1 ± 1.30
PLA + 1% Fe	24.3 ± 2.1	0.63 ± 0.06	3585 ± 186	70.8 ± 1.61
PLA + 5% Fe	26.4 ± 2.6	0.65 ± 0.08	3460 ± 146	67.5 ± 0.94
PLA + 10% Fe	30.0 ± 1.5	0.72 ± 0.09	3120 ± 169	66.8 ± 1.21
PLA + 0.1% nanoFe	32.6 ± 0.8	0.74 ± 0.08	3665 ± 170	67.2 ± 1.76
PLA + 0.5% nanoFe	35.5 ± 1.2	0.95 ± 0.10	3680 ± 125	65.1 ± 1.29
PLA + 1.0% nanoFe	43.6 ± 0.7	1.05 ± 0.06	3590 ± 210	63.8 ± 1.15
PLA + 1% nanoFe + 1% PEG (powder)	46.0 ± 0.8	1.60 ± 0.07	3425 ± 130	58.7 ± 3.81
PLA + 1% nanoFe + 1% PEG (fluid)	34.8 ± 1.2	1.20 ± 0.13	2901 ± 153	67.5 ± 2.52
PLA + 10% Fe + 20% PEG (fluid)	42.1 ± 2.1	1.35 ± 0.11	3235 ± 171	64.1 ± 4.04

**Table 4 polymers-17-00146-t004:** DSC results of PLA-based samples.

Sample	Tg			ΔH_c_ [J/g]	T_m_ [°C]	ΔH_m_ [J/g]	X_c_ [%]
	Onset [°C]	Midpoint [°C]	ΔCp [J/(°C·g)]				
Pure PLA	45.24	50.21	0.7318	3.42	153.25	19.91	17.60
PLA + 10% Fe	66.17	67.34	0.9784	29.52	170.11	59.37	31.85
PLA + 5% Fe	67.08	68.30	1.1420	30.39	171.95	54.20	25.41
PLA + 1% Fe	65.70	66.32	0.7750	32.42	169.69	51.32	20.71
PLA + 1.0% nanoFe	60.98	63.18	0.5109	14.16	169.53	56.23	44.89
PLA + 0.5% nanoFe	62.16	63.05	0.4432	12.49	169.68	53.58	43.85
PLA + 0.1% nanoFe	60.71	61.97	0.5382	26.99	169.96	50.44	25.02

**Table 5 polymers-17-00146-t005:** DSC results of PLA samples with PEG additives.

Sample	*Tg*			ΔH_c_ [J·g^−1^]	T_m_ [°C]	ΔH_m_ [J·g^−1^]	X_c_ [%]
	Onset [°C]	Midpoint [°C]	ΔCp [J·(°C·g)^−1^]				
PLA	45.24	50.21	0.7318	3.42	153.25	19.91	17.60
PLA + 1% nanoFe + 1% PEG (powder)	57.64	59.45	0.4681	11.17	167.52	56.12	47.97
PLA + 1% nanoFe + 1% PEG (fluid)	49.70	51.65	1.3000	19.08	164.22	55.36	38.72
PLA + 10% Fe + 20% PEG (fluid)	50.86	54.35	1.9710	17.70	164.89	57.29	42.25

## Data Availability

The original contributions presented in this study are included in the article. Further inquiries can be directed to the corresponding author.

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
