# Peer review of "The Influence of Iron Particles and Polyethylene Glycol on Selected Properties of Polylactide-Based Composites"

_polymers, 2025, doi:10.3390/polym17020146_

Round 1

Reviewer 1 Report

Comments and Suggestions for Authors

Oksiuta, et al. detailed study the effects of iron powder, nano-iron and nano-iron/PEG in the mechanical, thermal and degradation properties of PLA composites. The nano-iron performed better performance but lower mechanical properties. With combination nano-iron and PEG, the mechanical properties were improved. This work provides an effective method to improve the properties of PLA composites for medical applications. I suggest major revision before publishing.

1. In the abstract, the authors claim that the nano-iron performed better performance than iron powder. It should be detailed which performance/properties are better.

2. In the introduction part, it would be better to discuss the benefits of iron for PLA composites from the previous study, which is also the reason why the authors choose iron in this work.

3. Why 0.1%-1% nanoFe is studied but not higher ratio? The performance of the PLA/nanoFe continues increasing with more nanoFe. It will be beneficial to study higher ratios of nanoFe. Same for PEG. Only 1% and 20% of PEG was studied. A continuous ratio effect study is needed.

4. There is a lack of discussion about Young’s modules.

5. Morphology and structure study is needed to support the explanation of for the mass change in line 190-195. For example, during different days, an NMR spectrum and confocal microscope images should be taken to see the morphology change.

6. TGA is needed to study the degradation performance.

Author Response

  1. In the abstract, the authors claim that the nano-iron performed better performance than iron powder. It should be detailed which performance/properties are better.

The abstract outlines the superior properties of nanoiron composites in comparison to those comprising iron. These properties are tensile strength and elongation (text marked in red)..

  1. In the introduction part, it would be better to discuss the benefits of iron for PLA composites from the previous study, which is also the reason why the authors choose iron in this work.

The introduction sets out the rationale for the topic under discussion, building on the insights provided in the preceding article (text marked in red).

  1. Why 0.1%-1% nanoFe is studied but not higher ratio? The performance of the PLA/nanoFe continues increasing with more nanoFe. It will be beneficial to study higher ratios of nanoFe. Same for PEG. Only 1% and 20% of PEG was studied. A continuous ratio effect study is needed.

We would like to thank the reviewer for bringing this aspect to our attention. It was our preliminary work with nanopowder based on the literature available and the volume ratio. Also, for comparison it is equal to 1% of microFe powder.  In future work we will take this suggestion into account increasing the ratio of the nanopowder. It can be emphasized that indeed in the literature one can find numerous publications on the content of fillers up to several dozen percent, but also only up to 1%.

  1. There is a lack of discussion about Young’s modules.

We are grateful for your input and have incorporated a discussion on Young's modulus into the document (text in red).

  1. Morphology and structure study is needed to support the explanation of for the mass change in line 190-195. For example, during different days, an NMR spectrum and confocal microscope images should be taken to see the morphology change.

We are grateful for your valuable feedback on the article. Thank you for the insight and commitment you have demonstrated in the analysis of our text. However, we would like to inform you that we will take your suggestion into account in the next article, to further refine the issues raised.

  1. TGA is needed to study the degradation performance.

We sincerely appreciate your thoughtful feedback on our article. Please be assured that we will consider your suggestion in our next article, aiming to further enhance the discussion of the topics addressed. Some preliminary TGA results of the PLA+Fe+PEG composite have been investigated already (see Fig.1),  but it is out of the scope of the present paper.

Reviewer 2 Report

Comments and Suggestions for Authors

The research concerns the preparation and characterization of formulations based on polylactic acid containing iron powders, micro and nano, and polyethylene glycol (PEG). Specifically, all the materials, obtained by mixing in the molten state, have been characterized mainly in terms of thermal, mechanical properties, hardness and morphological aspects.

From the above it is clear that the research joins a myriad of scientific contributions already available in literature and aimed at reducing the intrinsic limitations of use of Polylactic acid.

Regarding further experimental determinations, the results of which are summarized in paragraph 3.2, however, the experimental procedures have not been reported in paragraph 2 nor is their objective clear.

Therefore, the authors are advised to integrate the text with details that justify the meaning of the tests in paragraph 3.2. Furthermore, specific attention is required for the following points.

-        In the abstract the authors state that the presence of nanometric iron particles improves the properties of the matrix compared to the presence of micrometric particles of the same additive while, in a subsequent sentence, they state that, regardless of the size of the iron particles, they degrade the mechanical properties. It is advisable to specify which properties are positively influenced by the presence of nano iron powder otherwise the consecutive statements are misleading for the reader.

-        In lines 158-160 it is written "Note, that both strengty and elongation of these composites increase with the increase in the content of nanoFe powder, from 0.1 wt.% up to 1.0 wt.%.". Probably the upper limit of iron nanoparticles content is 10 wt%. Please check and correct.

-        Table 4 and Table 5. The value of the degree of crystallinity of neat PLA (44.66%) is surprising. How was it evaluated? The indicated crystallinity value is atypical for PLA. Has it undergone annealing? If so, specify the annealing conditions.

In conclusion, MAJOR revisions to address the above mentioned aspects are recommended.

Author Response

Therefore, the authors are advised to integrate the text with details that justify the meaning of the tests in paragraph 3.2. Furthermore, specific attention is required for the following points.

Thank you for pointing this out. Relevant information was added in section 2.5 (marked in red in the text).

-        In the abstract the authors state that the presence of nanometric iron particles improves the properties of the matrix compared to the presence of micrometric particles of the same additive while, in a subsequent sentence, they state that, regardless of the size of the iron particles, they degrade the mechanical properties. It is advisable to specify which properties are positively influenced by the presence of nano iron powder otherwise the consecutive statements are misleading for the reader.

The abstract was expanded to include details of the specific mechanical properties of polylactide that were enhanced by the incorporation of iron powders and nano-iron (text marked in red)

-        In lines 158-160 it is written "Note, that both strengty and elongation of these composites increase with the increase in the content of nanoFe powder, from 0.1 wt.% up to 1.0 wt.%.". Probably the upper limit of iron nanoparticles content is 10 wt%. Please check and correct.

Thank you for your feedback and the dedication you have demonstrated in your analysis of our text. Typically the content of nanoparticles in the polymer matrix is 3-5%, but there is few literature data about lower nanoparticle content. Due to the fact that nanomaterials are expensive and also due to the unknown knowledge about the mechanism of nanoparticle migration into the human body, our approach was to limit the amount of nanoparticles up to 1.0%.

-        Table 4 and Table 5. The value of the degree of crystallinity of neat PLA (44.66%) is surprising. How was it evaluated? The indicated crystallinity value is atypical for PLA. Has it undergone annealing? If so, specify the annealing conditions.

Thank you very much for your remark. We have checked and recalculated the obtained result again and corrected the crystallinity of neat PLA is 17.60% and other samples. The incorrect data was corrected in the text (see Tables 4 and 5 marked in red).

Reviewer 3 Report

Comments and Suggestions for Authors This manuscript reported the Mechanical and Thermal Properties of Polylactide Composites Modified with Iron and Polyethylene Glycol Additives. Even though this article is fixed with the scope of our journal. However, there are some problems for authors to improve their article before acceptance.
1. The Abstract is too generic sentences, in order to attracted attention from readers please add scientific articles into this part i.e., some results of your study.
2. I agree with the aurthor’s expertise, but the author would be indicated and emphasis about new finding of this work when compared with your previous research work.
3. As mentioned about the effect of molecular weight on materials’ performance. How about the Mw of PLA (3001D) used in this study?
4. Did you investigate the effect of particle size of Fe on their mechanical properties? If yes, how about the result? If not, why?
5. The author mentioned “temperature of hopper zone 50℃,” Why it too low?
6. Degradation process test should be used TGA analysis, not only investigate the weight loss of the sample!!!!
7. Put some relative error in the result of Figure 4!!!
8. The mechanism of degradation is very important, the author should be provided.
9. How about the fracture surface of materials? It is should be investigated by SEM!!!
10. Degree of crystallinity of materials should be increased with increases in the filler concentration due to its acting as a nucleating agent within the composites. Why decreased?

Author Response

  1. The Abstract is too generic sentences, in order to attracted attention from readers please add scientific articles into this part i.e., some results of your study.

The abstract was revised according to the reviewers' recommendations (text marked in red).

  1. I agree with the aurthor’s expertise, but the author would be indicated and emphasis about new finding of this work when compared with your previous research work.

This work is a continuation of previous research [22] by Oksiuta et al. Polymers 2020, 12, 2939. Based on our previous experience, we changed the composition of the Fe content from 1.0 to 10wt.%, morphology of the iron powder, applying nano-iron particles with a range of 0.1 to 1.0wt.% and adding PEG as a compounder in liquid and powder form. Thus, we think that our new finding was completely different and presented in this manuscript.

  1. As mentioned about the effect of molecular weight on materials’ performance. How about the Mw of PLA (3001D) used in this study?

The molecular weight of the base material was not tested. It was not the subject of our consideration in this project. Nevertheless, we thank you for this comment. However, it is known that it has an impact on the mechanical properties and the rate of biodegradation of polylactide. The higher the molecular weight, the more favorable the mechanical properties and the longer the disintegration period. And vice versa. These issues will be discussed in a subsequent paper.

  1. Did you investigate the effect of particle size of Fe on their mechanical properties? If yes, how about the result? If not, why?

Thank you for this question. We did not investigate the effect of iron particle size on their mechanical properties, because we used commercially available powders. From the results however, indirectly, the influence of the coarse and fine fraction of iron particle size on the phenomena related to the bonding of powders can be observed at the sample surface fractures presented in Fig. 11 and 12. From the analysis of the obtained images, it can be concluded that in future work, a more homogeneous and smaller fraction of iron powders should be used.

  1. The author mentioned “temperature of hopper zone 50℃,” Why it too low?

Our experience shows that a temperature of 50°C in the hopper zone of the extruder is sufficient to carry out the extrusion process, and we have not been able to find information as to why it is too low. In our opinion, this temperature was determined correctly, based on the authors' experience, the thermal characteristics of PLA and the specific nature of the extrusion process.

  1. Degradation process test should be used TGA analysis, not only investigate the weight loss of the sample!!!!

Thank you for your suggestion. We will perform the degradation process with TGA analysis in the future study when the research work will be limited to a specific group of targeted material.

  1. Put some relative error in the result of Figure 4!!!

Standard deviations have been added to the graphs in Figures 4, 5 and 6.

  1. The mechanism of degradation is very important, the author should be provided.

The mechanism of degradation was discussed in lines 183-195 (before corrections), however due to the Reviver's suggestion this part was extended and the text was changed accordingly (marked in red).

  1. How about the fracture surface of materials? It is should be investigated by SEM!!!

We are grateful that you noticed this omission, we have posted SEM images (Figs.11 and 12) of the fracture surfaces of the tested materials with an appropriate comment (marked in red in the text).

  1. Degree of crystallinity of materials should be increased with increases in the filler concentration due to its acting as a nucleating agent within the composites.
    Why decreased?

Thank you for your valuable comment. For this purpose, the degree of crystallinity for neat PLA was re-examined and re-calculated and it turned out that the degree of crystallinity of neat PLA is 17.60%, much lower than for composite samples. Therefore, it was also recalculated and changed in the text and Tables. The incorrect data regarding PLA was the result of an error on our part. The text describing this issue has been changed accordingly.

Round 2

Reviewer 1 Report

Comments and Suggestions for Authors

The author has addressed most questions effectively. However, some concerns have been deferred to a subsequent paper. I recommend at least including TGA analysis in this paper to more comprehensively characterize the thermal properties.

Author Response

The author has addressed most questions effectively. However, some concerns have been deferred to a subsequent paper. I recommend at least including TGA analysis in this paper to more comprehensively characterize the thermal properties.

Thank you very much for taking the time to review this manuscript again. Please find our responses below and highlighted changes in the re-submitted files.

 2. Answer

We agree with the Reviewer’s comment, but at the moment we cannot perform full TGA analysis, due to the technical issue. The point is that the first studies were started almost a year ago and now we would have to prepare new (fresh) samples, which will significantly extend the time of publishing the manuscript. We will perform these TGA analysis, together with their description, in the next article in this series. PhD student Dominika Nowinska is also writing a dissertation on this topic. In addition, the title of the manuscript will be slightly change and instead of the word thermal properties we replaced it with the word selected properties, which is consistent with the content included in the article.

Reviewer 2 Report

Comments and Suggestions for Authors

The revised version of the manuscript can be accepted for publication.

Author Response

Thank you very much for taking the time to review this manuscript and your final approval.

Reviewer 3 Report

Comments and Suggestions for Authors

accepted

Author Response

(The authors gave the same response as above.)
